# An atlas of human kinase regulation

David Ochoa[1], Mindaugas Jonikas[2], Robert T Lawrence[3], Bachir El Debs[4], Joel Selkrig[4], Athanasios Typas[4], Judit Villén[3], Silvia DM Santos[2] & Pedro Beltrao[1,*]

## Abstract

The coordinated regulation of protein kinases is a rapid mechanism that integrates diverse cues and swiftly determines appropriate cellular responses. However, our understanding of cellular decision-making has been limited by the small number of simultaneously monitored phospho-regulatory events. Here, we have estimated changes in activity in 215 human kinases in 399 conditions derived from a large compilation of phosphopeptide quantifications. This atlas identifies commonly regulated kinases as those that are central in the signaling network and defines the logic relationships between kinase pairs. Co-regulation along the conditions predicts kinase–complex and kinase–substrate associations. Additionally, the kinase regulation profile acts as a molecular fingerprint to identify related and opposing signaling states. Using this atlas, we identified essential mediators of stem cell differentiation, modulators of *Salmonella* infection, and new targets of AKT1. This provides a global view of human phosphorylation-based signaling and the necessary context to better understand kinase-driven decision-making.

**Keywords** cell fate; human; kinase activity; phosphoproteomics; signaling
**Subject Categories** Genome-Scale & Integrative Biology; Post-translational Modifications, Proteolysis & Proteomics; Signal Transduction
**Mol Syst Biol. (2016) 12: 888**

## Introduction

Cells need to constantly adapt to internal and external conditions in order to maintain homoeostasis. During cellular decision-making, signal transduction networks dynamically change in response to cues, triggering cellular state-defining responses. Multiple mechanisms exist to transfer this information from sensors to the corresponding molecular responses, one of the fastest being the reversible posttranslational modification of proteins (PTMs). Through these targeted modifications, such as phosphorylation, the cell can quickly alter enzymatic activities, protein interactions, or subcellular localization in order to produce a coordinated response to a given stimulus (Pawson, 2004). Protein phospho-regulation constitutes a highly conserved regulatory mechanism relevant for a broad set of biological functions and diseases (Beltrao *et al*, 2012).

Over the past decades, our view of cellular signaling has advanced from an idea of isolated and linear cascades to highly complex and cooperative regulatory networks (Jordan *et al*, 2000; Gibson, 2009). Different perturbations in cellular conditions often activate different sets of interconnected kinases, ultimately triggering appropriate cellular responses. The complete understanding of such cell fate decisions would require the systematic measurement of changes in kinase activities under multiple perturbations, but the small number of quantified regulatory events (i.e. tens) that were possible to date has limited our knowledge of cellular decision-making and its molecular consequences (Garmaroudi *et al*, 2010; Bendall *et al*, 2011; Kim *et al*, 2011; Niepel *et al*, 2013).

Advances in mass spectrometry and enrichment methods now allow measuring changes in thousands of phosphorylated peptides at a very high temporal resolution (Olsen & Mann, 2013; Humphrey *et al*, 2015; Kanshin *et al*, 2015). Recent studies on human quantitative phosphorylation include responses at different cell cycle stages (Dephoure *et al*, 2008; Olsen *et al*, 2010), after DNA damage (Beli *et al*, 2012), EGF stimulation (Olsen *et al*, 2006; Mertins *et al*, 2012), prostaglandin stimulation (de Graaf *et al*, 2014) and different kinase inhibitions (Hsu *et al*, 2011; Kettenbach *et al*, 2011; Weber *et al*, 2012; Oppermann *et al*, 2013) among many others. More recently, improvements in experimental and computational methods have fostered the study of differential regulation of phosphosites and kinases in different cancer types (Casado *et al*, 2013), the modeling of drug resistance (Wilkes *et al*, 2015) and inference of more precise pathway models (Terfve *et al*, 2015). We suggest that the integrated analysis of phosphoproteomic responses after a wide panel of heterogeneous perturbations can expedite our understanding of cell decision-making processes.

In this study, we have compiled condition-dependent changes in human protein phosphorylation derived from 2,940,379 phosphopeptide quantifications in 435 heterogeneous perturbations. After quality control and normalization, we infer and benchmark the changes in 215 kinase activities in 399 conditions. We show that the similarity of kinase regulatory profiles can be used as a fingerprint to compare conditions in order to, for example, identify perturbations

1 European Molecular Biology Laboratory, European Bioinformatics Institute (EMBL-EBI), Hinxton, UK
2 Quantitative Cell Biology Group, MRC Clinical Sciences Centre, Imperial College, London, UK
3 Department of Genome Sciences, University of Washington, Seattle, WA,USA
4 Genome Biology Unit, European Molecular Biology Laboratory (EMBL), Heidelberg, Germany
*Corresponding author. Tel: +44 1223 494 610; E-mail: pbeltrao@ebi.ac.uk

that modulate the kinase activity changes of a condition of interest. The large number of conditions analyzed allowed us to identify the kinases that are broad regulators (i.e. generalist kinases), found to be central kinases of the signaling network. Individual kinase profiles across conditions were also informative to recapitulate known kinase–kinase interactions and to identify novel co-regulated complexes and phosphosites acting as potential effectors.

# Results

**Landscape of kinase activity responses under perturbation**

To extensively study the heterogeneity and specificity of the human signaling response, we compiled and standardized 41 quantitative studies reporting the relative changes in phosphopeptide abundance after perturbation (see Materials and Methods). From the detected peptides, we collected identifications for 119,710 phosphosites in 12,505 proteins, 63% of which were already reported in phosphosite databases (Fig EV1). For these sites, we normalized a total of 2,940,379 quantitative changes in phosphopeptide abundance in a panel of 435 biological conditions covering a broad spectrum of perturbations including targeted kinase inhibition, induced hESC differentiation, or cell cycle progression, among many others (Appendix Fig S1, Table EV1). Only 1% of all phosphorylated sites were reported in more than 60% of the studies, whereas 52% of the sites were quantified in one single study (Fig EV1). The observed data sparsity, a common problem in shotgun proteomics, is frequently derived from the accumulation of technical and biological variants. To prevent the aggregation of false positives, only phosphosites observed in two or more independent studies were considered in downstream analysis.

In order to circumvent the problem of incomplete coverage due to the different sets of quantified sites in each study, we avoid analyzing individual phosphosites. Instead, we predicted the changes in kinase activity by testing the enrichment on differentially regulated phosphosites among the known substrates of each kinase (Fig 1A). Using a modified version of the weighted kinase set enrichment analysis (KSEA) (Subramanian *et al*, 2005; Casado *et al*, 2013), we quantified the regulation of 215 kinases in a range between 10 and 399 perturbations (Fig 1A, Table EV2, Materials and Methods). To verify the ability of KSEA to quantitatively measure the changes in kinase regulation, we performed a series of benchmark tests based on prior knowledge.

The known mechanism of action of certain biological processes or compounds suggests different perturbations in which specific kinase regulation is expected. For instance, the ATM (ataxia telangiectasia, mutated) and ATR (Ataxia Telangiectasia and Rad3-related) kinases display direct regulation corroborated by the KSEA activities under DNA damaging conditions (Fig 1B). Similarly, the kinases in the MAPK/Erk pathway accurately display activation 10 min after EGF stimulation (Fig 1B). Conversely, the KSEA estimates also report decrease in kinase activities as in the case of the epidermal growth factor receptor (EGFR) inhibition mediated by erlotinib and gefitinib or mTOR inhibition by Torin1 (Fig 1B). Overall, the KSEA activity shows predictive power to discriminate expected regulation in 132 kinase–condition pairs (Fig 1C, Table EV3, area under the ROC curve = 0.75).

To further validate the kinase regulation inference, we compared the KSEA activities across conditions with the corresponding changes in kinase regulatory phosphosites collected in the atlas. For example, phosphorylation of the activation loop residue threonine 287 (T287), known to result in an increased catalytic activity of AURKA, presents a significant co-regulation with the AURKA KSEA activity (Spearman's $\rho = 0.6$, $P = 0.02$). Phosphorylation of T287 and KSEA activity derived from AURKA substrates are both decreased as AURKA inhibitor MLN8054 concentration increases (Fig 1D). Overall, we observe a significantly higher correlation between the KSEA activities and the changes in kinase auto-regulatory sites (Student's $t$-test, $P = 1.7 \times 10^{-4}$) (Fig 1E). Finally, we compared the kinase regulation with those assayed in a previous study using phospho-specific antibodies under similar conditions (Hill *et al*, 2016). As an example, the KSEA activities 10 min after EGF stimulation significantly correlate ($\rho = 0.53$, $P = 0.008$) with the antibody-based quantified phospho-regulation 15 min after EGF stimulation (Fig 1F). Despite the differences of both assays, the profile of inferred changes based on 26 phospho-dependent antibodies and the MS-based KSEA activities for the equivalent kinases present significantly higher correlations when cells are stimulated with similar EGFR activating conditions (Fig 1G, Student's one-sided $t$-test $P = 2.7 \times 10^{-5}$, Appendix Fig S2).

Together, these results not only validate the activity inference for individual kinases but strongly suggest the profile of kinase activity changes can serve as molecular barcodes of the cellular signaling state.

**Inhibition of inferred regulatory kinases impairs state transition during PMA-induced hESC differentiation**

To further validate the inferred KSEA activities, we experimentally measured the activity changes of 10 kinases using immunohistochemistry (Table EV4) during human embryonic stem cell (hESC) differentiation induced by Phorbol 12-myristate 13-acetate (PMA), a perturbation compiled in the phosphoproteomic atlas (Rigbolt *et al*, 2011; Fig 2A and B). Immunofluorescence and KSEA substrate-derived activities 30 and 60 min after PMA treatment agree in their regulatory effect—activatory or inhibitory—for 14 out of the 20 quantifications (Figs 2C and EV2). Several of the concordant changes are expected to occur during differentiation such as for PKC (PRKCA) (Feng *et al*, 2012), Erk2, RSK (RPS6KA1), GSK3A, and GSK3B (Kinehara *et al*, 2013). For CDK1, the predicted activities were corroborated using an antibody targeting cyclin B1 pS126 (CycB1/CCNB1), a phosphorylation required for the activation of the CDK1-cyclin B1 complex.

Not all regulated kinases may be functional relevant for the process under study. To discriminate driver regulatory kinases from secondary kinases activated as a consequence of the differentiation process, we monitored the PMA-induced transition in the presence of kinase inhibitors (Table EV5). Using immunofluorescence, we quantified the cytoplasmic abundance of Oct4 and Erg1 as respective early and late markers of PMA-driven differentiation (Niwa *et al*, 2000; Kinehara *et al*, 2014; Fig EV3, Appendix Fig S3). Interestingly, Erk2 inhibition induced the strongest disruption of Erg1 expression. Inhibition of CDK1 also appears to delay the increase in Erg1 expression and, potentially, the differentiation process. On the other hand, the inhibition of RSK (RPS6KA1) shows the strongest

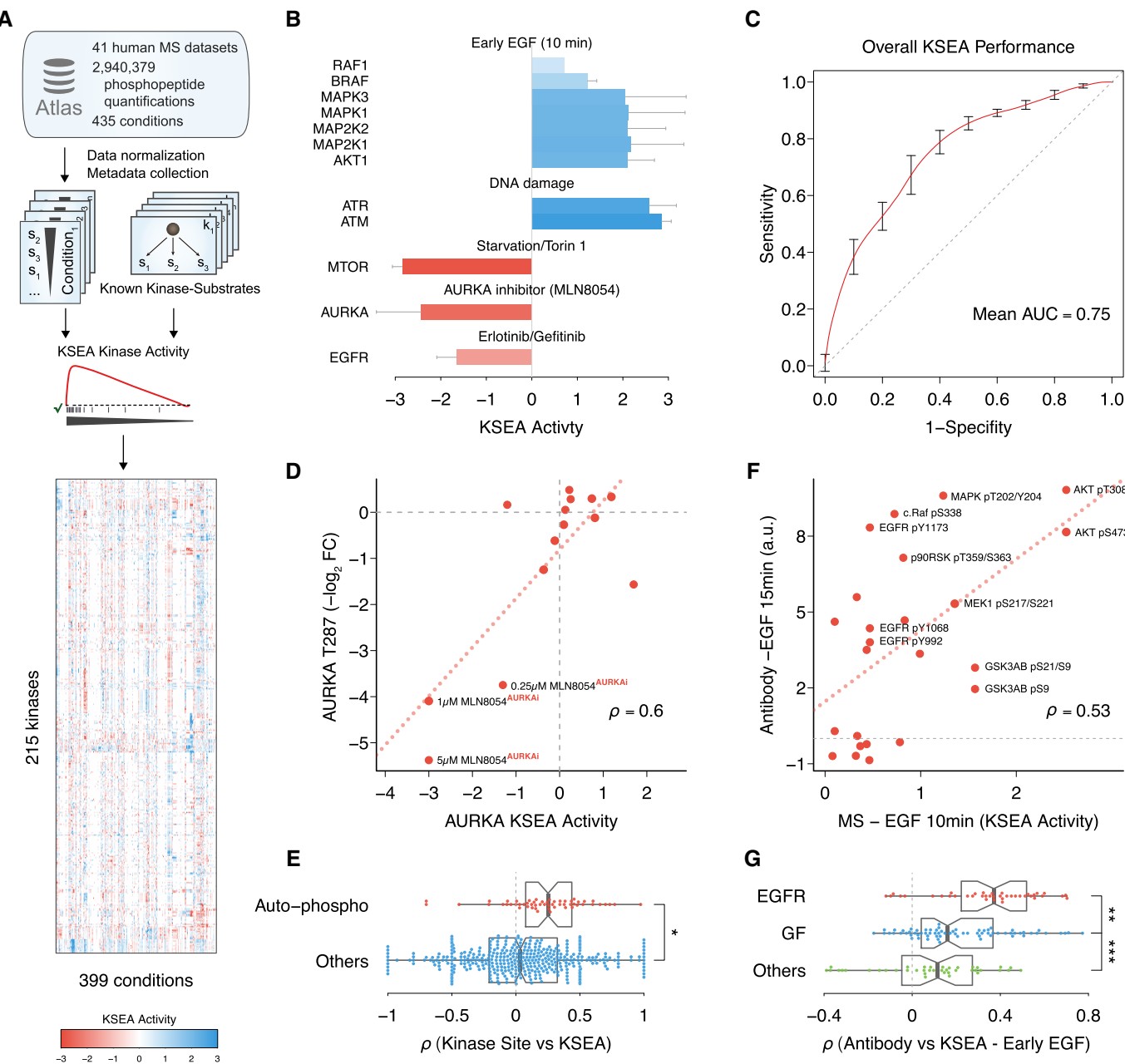

**Figure 1.   Kinome-wide activity regulation derived from known substrates and 41 quantitative phosphoproteomic studies.**

A   Schematic of the data compilation effort and kinase activity prediction using Kinase Set Enrichment Analysis (KSEA).

B   Expected kinase response after activation or inhibition. When available ($n \geq 2$), error bars represent standard deviation over the mean KSEA activity.

C   Receiver operating characteristic (ROC) representing the discriminative power of the KSEA activity to separate a set of 132 kinase–condition pairs expected to display regulation. As negatives, 1,000 random sets were generated containing the same number of kinase–condition pairs from the same set of kinases and conditions. Curve displays average ROC curve and bars standard deviation. AUC, area under the ROC curve.

D   Regression analysis between Aurora kinase A (AURKA) regulatory site T287 and AURKA KSEA activity across all quantified conditions. Labeled conditions correspond to different concentrations of the AURKA inhibitor MLN8054 under mitosis.

E   Comparison between Spearman correlation coefficients obtained between KSEA-inferred kinase activities, quantifications of regulatory sites susceptible of auto-phosphorylation ($n = 56$), or other regulatory sites in kinases ($n = 395$). The boxes represent the 1st, 2nd (median) and 3rd quartiles and the whiskers indicate 1.5 times the interquartile range (IQR). Two-sided Student's $t$-test *$P = 1.7 \times 10^{-4}$.

F   Linear regression between KSEA activities 10 min after EGF stimulation and activities measured with RPPA targeting regulatory phosphorylations 15 min after adding EGF.

G   Spearman correlation coefficients between the profile of 24 kinase activities estimated with KSEA 10 min after EGF stimulation ($n = 40$) and the activities of the same kinases measured with RPPA after stimulating the cell with different ligands. EGFR ligands, EGF or NRG1; other growth factors (GF), HGF, IGF, insulin, or FGF ($n = 70$); or control conditions, serum or PBS ($n = 40$). The boxes represent the 1st, 2nd (median) and 3rd quartiles and the whiskers indicate 1.5 times the IQR. Two-sided Student's $t$-test **$P = 0.005$. ***$P = 0.004$.

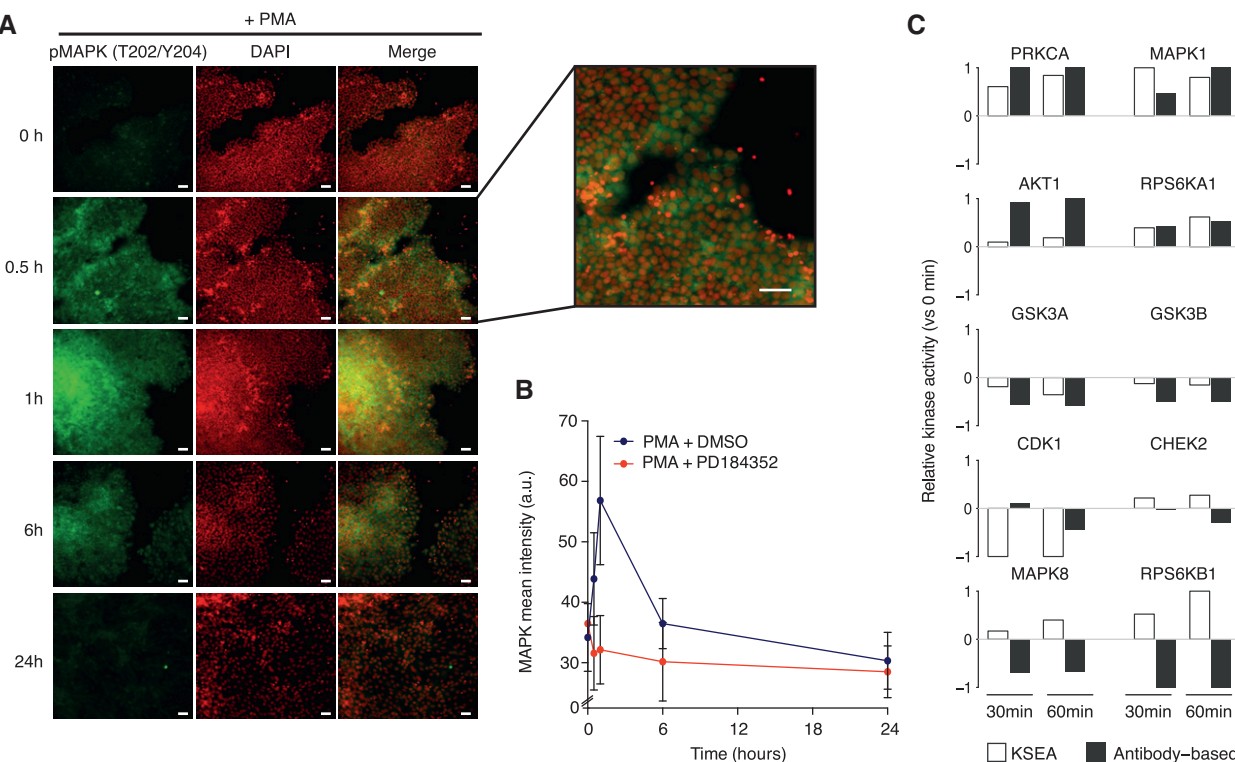

**Figure 2. Inhibition of inferred regulatory kinases impairs PMA-induced differentiation of hESC.**

A  Representative images of differentiation marker MAPK (pT202/Y204) expression in hESCs stimulated with PMA. Scale bars: 30 μm.

B  Time course quantification of MAPK activation levels after PMA stimulation in the presence or absence of MAPK inhibitor (PD184352). Bars represent mean ± SD (n > 1,000).

C  Relative changes in kinase activities using Kinase Set Enrichment Analysis (KSEA) benchmarked against antibody-measured reporter phosphosites in the intervals 0–30 and 0–60 min.

induction of Erg1 expression after treatment, suggesting a possible role of its activity in the maintenance of the pluripotent state. The inhibition of GSK and S6 (RPS6KB1) kinases results in a small increase in PMA-induced Erg1 expression only during the early time points. Overall, these results show how the KSEA-based inference can predict regulated kinases and therefore predict those that are more likely to be functionally relevant in specific conditions. This illustrates how the kinase atlas can serve as a useful cell signaling resource.

**Kinase regulation profiles as molecular fingerprints of cellular signaling states**

The diversity of the compiled perturbations as well as the extent of the kinases for which regulation is inferred constitutes a resource to study fundamental aspects of cell signaling. To demonstrate that the biological variation dominates over the technical variation, we tested whether related kinases display co-regulation across conditions and, similarly, related conditions show similar patterns of kinase regulation. We observed that significant correlations between the KSEA activities were more frequent between kinases one or two steps away in the pathway than between those farther away (Fig EV4A). This observation remains true when excluding kinase pairs sharing substrate sites (Fig EV4B). Similarly, we confirmed that pairs of related conditions measured in different studies tend to

have similar profiles of KSEA activities (Fig EV4C). Furthermore, the correlation of kinase regulatory profiles is a very strong predictor of related conditions assayed in different studies (Fig EV4D, AUC = 0.93), but not of pairs of conditions from studies conducted in the same laboratory (AUC = 0.499), with the same cell line (AUC = 0.546) or with the same labeling method (AUC = 0.475). These results strongly suggest that the variation in kinase activities across conditions is primarily driven by biological effects rather than technical variation.

In order to explore the space of different signaling responses, we performed a principal component analysis (PCA) using the kinase regulation profiles derived from 58 well-characterized kinases (Fig 3A, Materials and Methods, Appendix Fig S4). The first two components separate related EGF conditions based on their expected signaling similarities and opposite to the EGFR pathway inhibitors (Fig 3A, symbols). The separation of perturbations in the reduced space is again independent of the publication of origin, reflecting instead the similarities in the signaling response. The systematic exploration of conditions in the reduced space also allows us to investigate commonalities in the decision-making process. Kinase loadings driving the sample separation in the PCA space reflect systematic differences on the regulation of different kinases (Appendix Fig S4C). In this way, we can identify different types of kinase logic relationships that apply to nearly all conditions (Fig 3B). Some kinase pairs are co-regulated—such as BRAF and

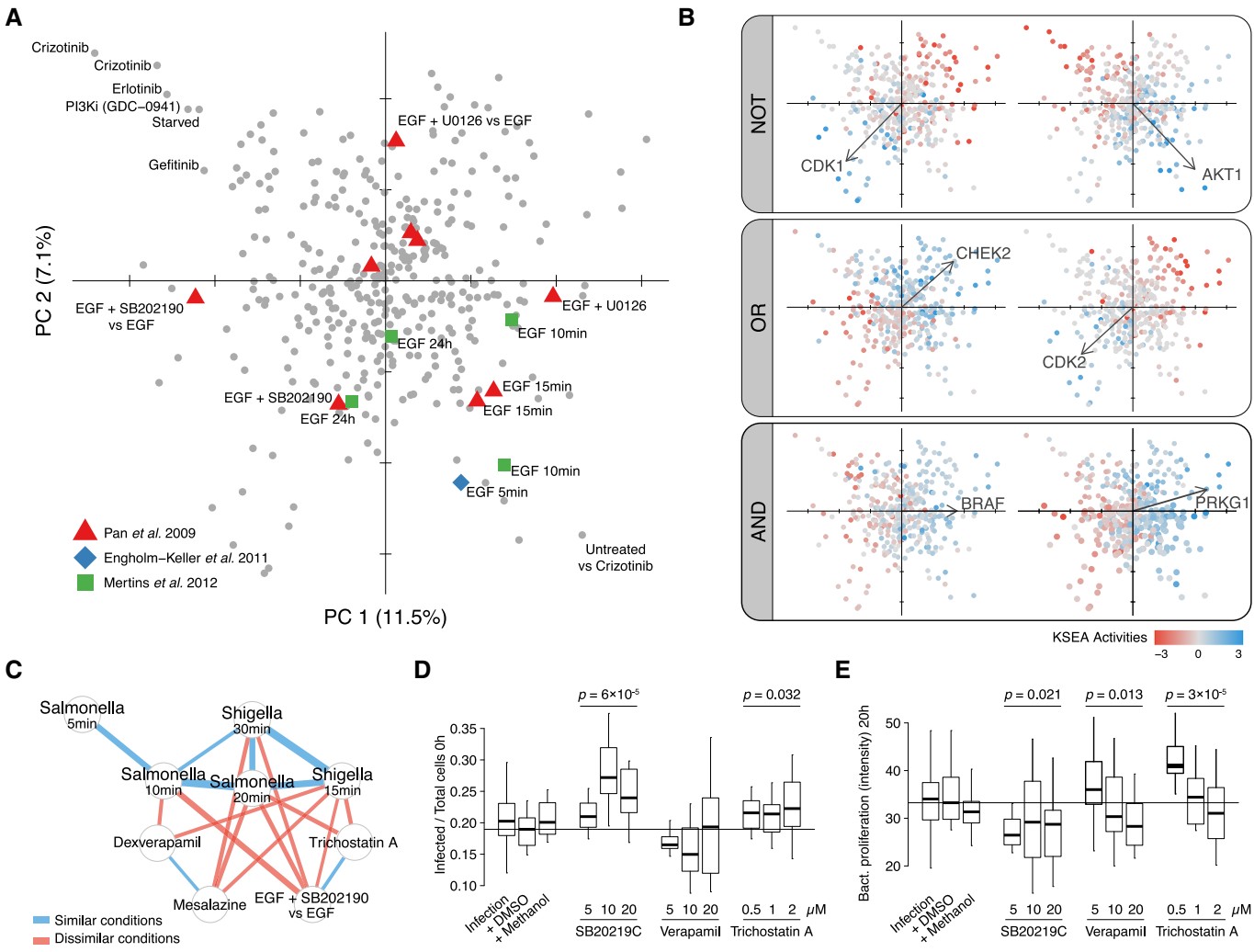

**Figure 3. Kinase activity profiles as fingerprints of the cell signaling state.**

A    Perturbation scores on the first two PCA components based on KSEA activity profiles of 52 well-characterized kinases. Symbols represent EGF-related perturbations in different studies.

B    Boolean logic relationships between kinase responses. Samples in two first components are colored by different KSEA activities. Vectors display kinase loadings.

C    Network displays significantly correlated or anti-correlated conditions in the context of early responses after bacterial infection. The strength of the correlations (blue) and anti-correlations (red) is displayed as the edge width.

D, E  Infection rate at 0 h (D) and bacterial proliferation after 20 h (E) when adding different concentrations of compounds displaying anti-correlated KSEA activity profiles with early responses after bacterial infection (4 biological replicates). Displayed significant ANOVA *P*-values evaluate differences between three drug concentrations and the DMSO control. The horizontal lines represent the median baseline value for the Infection + DMSO control.

PRKG1 (Fig 3B, AND)—or anti-correlated—such as CDK2 and CHEK (Fig 3B, OR). Alternatively, we also identify pairs of kinases that display exclusive regulation, whereby one is never regulated at the same time as the other. For example, AKT1 is regulated when CDK1 is not and vice versa (Fig 3B, NOT).

The results above show how extreme similarities or dissimilarities between profiles of activity changes facilitate the interpretation and generate hypothesis about the signaling in specific conditions. For example, perturbations under DNA damaging conditions display similar KSEA activity profiles that can be summarized as a signature of marker kinases (Fig EV5). Among the most similar conditions to two DNA damage conditions (ionizing radiation and etoposide) are compounds that are known to cause DNA damage and a sample

under G1-S transition obtained using a thymidine block that likely resulted in DNA damage. Conversely, cells treated with the inhibitor VE-821 targeting the DNA damage response kinase ATR show changes in activities anti-correlated with DNA damaging conditions (Fig EV5C). Therefore, kinase regulatory profiles can be used to identify perturbations that may mimic or modulate the kinase regulation occurring in a condition of interest. We further explored this notion in the study of two related *Shigella* and *Salmonella* infection states (Fig 3E). Among the anti-correlated conditions are 4 compounds that could potentially interfere with the infection process or the host response: SB202190 (p38 MAP kinase inhibitor), mesalazine (anti-inflammatory), trichostatin A (HDAC inhibitor), and verapamil (an efflux pump inhibitor). To validate the effect of

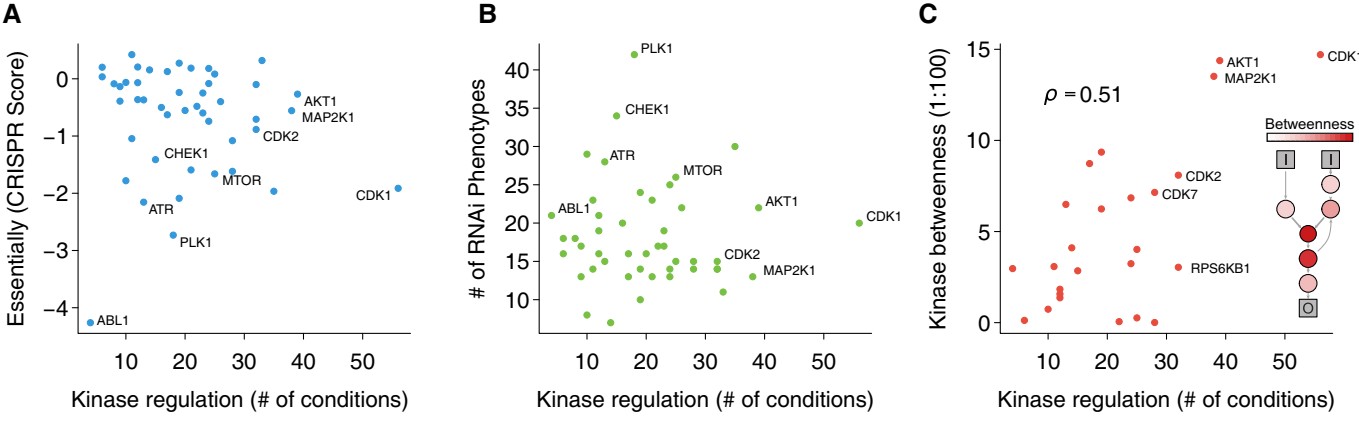

**Figure 4.  Relevance of generalist or specialist kinases.**

A  Genetic relevance of generalist and specialist kinases. Number of conditions where the kinase is regulated (absolute estimated kinase activity > 1.75) for each kinase with more than 10 known substrates against the depletion score from CRISPR essentiality screen (Wang *et al*, 2015). A lower depletion score is indicative of kinases that cause severe fitness defects when knocked-out.

B  Same number of conditions in which a kinase is regulated against the number of phenotypes shown by the knocked-down kinase (from a compilation of RNAi screens www.genomernai.org).

C  Same number of conditions in which a kinase is regulated against kinase centrality (betweenness) in signaling network. In the inner panel, a diagram illustrates the relationship between betweenness and the signaling network connectivity. Generalist kinases with more than 10 known substrates tend to have also high betweenness scores (Spearman's $\rho = 0.506$, $P = 9.8 \times 10^{-3}$). Kinases without shortest paths going through them were excluded.

these compounds, we have measured their impact on the invasion and proliferation of *Salmonella enterica* serotype Typhimurium (STm) in human cells (Fig 3D, Materials and Methods). Mesalazine showed no effect on either invasion or proliferation (data not shown). Trichostatin A and higher doses of SB202190 tend to promote invasion. SB202190 showed a consistent decrease in long-term STm proliferation while trichostatin A showed a trend for increase in STm proliferation that was clearer for lower doses of the drug. Verapamil had a significant effect on proliferation that was not consistent across different concentrations. These results show how modulators of signaling states of interest can be identified by comparing kinase regulatory profiles found in the atlas.

### Activity signatures reveal generalist and specialist kinases

The large panel of estimated changes in kinase activity across conditions allows us to classify kinases according to their degree of specificity. Some kinases, such as AKT or CDK1, are very often regulated across all conditions and can be defined as generalist kinases. Other kinases such as ATM and ATR are more narrowly regulated and can be considered specialist kinases. To study these two classes of kinases, we correlated the number of conditions in which kinases show changes in activity with the functional importance of the kinases measured in genetic experiments. Functional importance was scored as either the degree of essentiality from a CRISPR screen (Wang *et al*, 2015) or by the number of phenotypes from a compilation of RNAi screens (from www.genomernai.org) (Fig 4A and B). Kinases that have changes in activity in many conditions (e.g. generalist kinases) are not more likely to be functionally important than specialist kinases. For example, ATR or PLK1 are regulated in few conditions but tend to be essential. We observed however that generalist kinases, such as AKT and CDK1, are more central in the kinase signaling network as measured by the number of shortest

paths that traverse them in the directed kinase–kinase network (Fig 4C, $\rho = 0.506$, $P = 9.8 \times 10^{-3}$, excluding kinases with 0 betweenness). Kinases that are often regulated tend to occupy positions in the network where signaling is very likely to flow through based on the wiring of the network. Understanding the properties of generalist and specialist kinases may allow us to better understand the specificity of the signaling response, as well as to propose novel therapeutic targets and inform on the potential consequences of kinase inhibition.

### Kinase co-regulation identifies novel molecular effectors

The conditional depth of the kinase regulation atlas facilitates the search for co-regulated kinases and potential molecular effectors. Protein complexes are common signaling effectors that often display coordinated phospho-regulation with regulatory kinases. To search for kinase–complex co-regulation, we quantified the enrichment of regulated phosphosites within stable human complexes. We then correlate this enrichment with the KSEA activities across the panel of biological perturbations (Materials and Methods). Kinase–complex associations were validated if at least one subunit in the complex was a known substrate of the kinase. Overall, we found a very strong enrichment for known kinase targets among the kinase–complex associations predicted from co-regulation (Fig 5A, Table EV6). Using CDK1 as an example, we found a significant number of co-regulated complexes validated as direct substrates of CDK1 based on previous evidence, even though the actual substrate sites in the complex were not used to predict their association (Fig 5B). We have also identified examples of complex subunits functionally related to CDK1, but with no evidence yet of direct regulation. The chromatin assembly complex (CAF-1 complex), for instance, delivers newly synthesized H3/H4 dimers to the replication fork during the DNA synthesis (S) phase, shifting to secondary

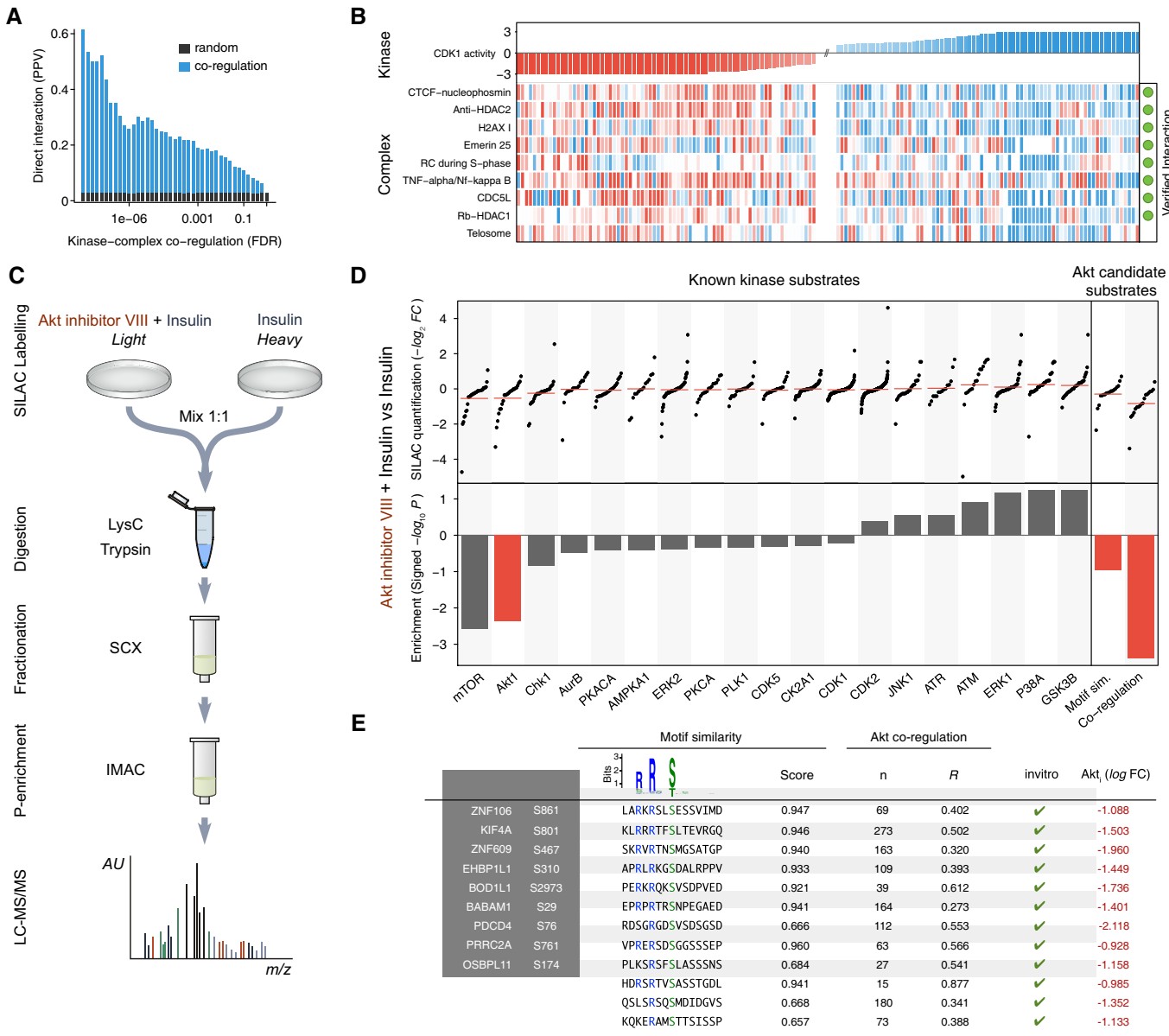

**Figure 5. Kinase co-regulation reveals candidate molecular effectors.**

A   Systematic evaluation of the kinase–complex associations based on the known direct interactions between kinases and complexes. The positive predictive value (PPV) is displayed against the false discovery rate (FDR). The baseline random expectation (in gray) represents the PPV of a random predictor trying to estimate associations between kinases and complexes.

B   Protein complexes showing correlated phospho-regulation with the activity of CDK1. The complexes marked in green contain at least one substrate of CDK1. Only the top correlated complexes are shown for the sake of clarity. Missing activities are displayed in white.

C   Experimental workflow to study phosphoproteome dynamics under AKT (AKT1) inhibition in insulin-stimulated HeLa cells.

D   Quantification of known kinase substrates after AKT inhibition of insulin-stimulated cells for all kinases with at least 14 known sites (top left) and their respective KSEA enrichment after 10,000 permutations (bottom left). Regulation under AKT inhibition of the top 24 unknown sites (number of quantified AKT known substrates) ranked based on their motif similarity, co-regulation with the known substrates or the combination of both (top right) and their corresponding enrichment on regulated sites after inhibition (bottom right).

E   List of high-confidence AKT substrates fulfilling the following criteria: down-regulation on AKT inhibition log$_2$ L/H < −0.9, positive co-regulation $P < 0.01$, motif similarity log-weights > 0.8, $mss > 0.6$, and all sites reported as *in vitro* substrates of AKT (Imamura *et al*, 2014).

functions during other stages of the cell cycle (Volk & Crispino, 2015). Although no specific site in the complex has been validated as CDK1 substrate, the observed co-regulation of CAF-1 and CDK1 ($r = 0.27$, FDR = $5 \times 10^{-4}$) was partially validated *in vitro* as CDK

inhibition prevents the replication-dependent nucleosome assembly in human cell extracts (Keller & Krude, 2000).

As an additional application of this approach, we tested whether co-regulation can also be predictive of novel AKT1 kinase target

sites. In order to validate these predictions, we measured the phosphorylation changes of 15,255 phosphosites in insulin-stimulated HeLa cells in the presence or absence of the AKT inhibitor VIII (Fig 5C, Table EV7). As expected, previously known AKT targets are, on average, down-regulated in the presence of the inhibitor (Fig 5D). Additionally, the substrates of downstream related kinases, such as mTOR or GSK, are also regulated. When predicting the same number of AKT targets as either sites strongly matching the AKT sequence preference or sites showing the most significant co-regulation with AKT across conditions, the latter showed much stronger down-regulation after AKT inhibition (Fig 5D).

In order to propose a list of new *bona fide* AKT targets, we short-listed those that are strongly co-regulated with the AKT activity ($P < 0.01$), match the AKT sequence preference, are down-regulated after AKT inhibition ($\log_2 L/H < -0.9$), and were also reported as *in vitro* AKT substrate sites (Imamura *et al*, 2014). We identified 12 AKT target phosphosites matching these stringent criteria (Fig 5E, Table EV8). S588 of TBC1D4 was previously described to be a potential AKT target sites, despite it not being present in our training set (Berwick *et al*, 2002; Kane *et al*, 2002). The tumor suppressor PDCD4 has also been shown to be regulated by AKT but at position S67 controlling its localization (Palamarchuk *et al*, 2005). We identified S76 as an AKT target, a residue important for PDCD4 degradation (Galan *et al*, 2014; Matsuhashi *et al*, 2014) that is phosphorylated under EGF stimulation (Matsuhashi *et al*, 2014). This suggests that AKT activity may directly control the protein levels of PDCD4. Another interesting target site is S801 of the kinesin KIF4A, a motor protein involved in the control of microtubule stability. The nearby position T799 is a target of Aurora B and the double alanine mutant T799A/S801A has impaired function (Nunes *et al*, 2013). In addition, both AKT and KIF4A regulate microtubule stability during cell migration (Onishi *et al*, 2007; Morris *et al*, 2014). Taken together, these results suggest that AKT directly targets KIF4A S801 and that this interaction plays a role in microtubule stabilization during migration.

# Discussion

Changes in internal or external conditions are sensed by cells which rapidly make decisions on how to mount an appropriate response. Difficulties in measuring activities and downstream consequences for a large number of kinases have limited the comprehensive understanding of the decision-making process. Here, we have compiled the changes in the human phosphoproteome across 399 perturbations, in order to create an atlas of regulation for 215 kinases, as well as the phospho-regulatory status of hundreds of effector protein complexes.

One of the main limitations preventing the integration of MS quantitative data has been the stochasticity of the peptide discovery. In this study, we have demonstrated how the analysis of sets of phosphosites (e.g. all kinase substrates) can help overcome the lack of coverage on phosphosite quantifications. The enrichment on biologically meaningful sets of sites also increases the robustness against technical variability, mostly due to differences in experimental protocols and analysis pipelines. Several results suggest the variation in kinase activities along conditions in our compendium is driven by biological factors: Related conditions have similar profiles

of kinase regulation (Fig EV4C and D); pairs of related kinases are co-regulated across the conditions (Fig EV4A and B); and co-regulation between kinase activity and effectors is predictive of kinase–target interactions (Fig 5). These results strongly indicate that this compendium constitutes a resource to identify novel biological associations. The measured changes in phosphorylation were not normalized for the changes in total protein abundance for the conditions compiled for this study. In particular, for longer time-scales this could be a confounding effect that should also be mitigated by the analysis of sets of sites. The described analysis can be periodically updated as new kinase targets are characterized and new human perturbation experiments refine the map of signaling responses. As such, we will maintain a growing online application (http://phosfate.com), where the community can explore the signaling regulation of all conditions in the atlas and analyze and compare their own phosphoproteomic datasets.

The larger number of perturbations identifies the possible roles of kinases and how they relate to each other. We have shown that generalist kinases do not tend to be more essential than specialist kinases but, instead, tend to be more central in the signaling network. One interpretation would be that generalist kinases are often regulated simply because they are located in the network where signals flow through more often. Recent phosphoproteomic studies of high temporal resolution have indicated that changes in phosphorylation occur in a timescale of seconds (Humphrey *et al*, 2015; Kanshin *et al*, 2015). Some fraction of the later changes in phosphorylation may be non-functional and potentially be consequence of the signaling propagating through the kinase signaling network (Kanshin *et al*, 2015). An alternative hypothesis could be that generalist kinases, central to the signaling network, can be robust to perturbation and the signal modulated by other kinases.

By comparing kinase and complex regulation in different signaling contexts, we have shown tight co-regulation between regulators and effectors that served to prioritize candidate interactions. However, regulation of substrate phosphosites or protein complexes is just one of the several potential consequences of a signaling response. Given the ease in collecting cellular phenotypes or large-scale biological measurements such as gene expression and metabolites, we propose this approach can be generalized to globally study the diversity of cellular molecular states. Additionally, as data from specific genetic backgrounds (e.g. cancer cell lines or primary tumors) become available, this research could help to interpret the actionable signaling consequences derived from specific sets of mutations, drawing a much more complete diagnostic of the cellular state.

# Materials and Methods

### Compilation of human quantitative phosphoproteomic data

The atlas integrates a compilation of 41 selected publications reporting human MS-derived changes in phosphopeptide abundance under 435 perturbations. In order to consider a dataset for the atlas, we required peptide sequences, phosphorylation identifications, and detailed description of the biological perturbation and control for a minimum of 1,000 phosphopeptides (Table EV1). From the 41

studies included, only the raw spectra derived from 16 studies were made available in public repositories. Three additional studies were deposited in the no longer available Tranche. The different labeling methodologies make difficult the processing of the public spectra with a common pipeline. Additionally, due to the poor metadata annotation of some of the conditions, reprocessing the spectra would not be possible for some of the datasets. For these reasons, we relied on the original MS identifications performed by the experimental groups. We collected all quantifications from the supplementary data available in each of the 41 studies, including both significant and not significant changes. To standardize the peptide sequences and site positions across studies, we mapped all peptides to the same reference proteome using Ensembl v73 (Cunningham *et al*, 2015). For data storage purposes, we mapped the peptide quantifications to all matching protein isoforms and transformed all quantifications into $\log_2$ ratios. Finally, the remapped 2,940,379 phosphopeptide quantifications and all their metadata were stored in a MySQL database that is publicly available in the Downloads section of http://phosfate.com.

### Data preprocessing and normalization

To standardize and further compare the datasets, we applied a series of quality control criteria. (i) We restricted the analysis to peptides mapped in the Ensembl canonical transcripts, resulting in 12,427 canonical phospho-modified proteins. According to Ensembl, the canonical transcripts correspond to the longest Consensus CDS (CCDS) model in each gene (Pruitt *et al*, 2009). (ii) The quantified changes of peptides with the same set of modifications but different sequences were merged into one single entry, therefore increasing the conditional coverage. On those cases where more than one quantified peptide contains the same set of modifications for the same condition, their ratios were averaged. (iii) The conditional ratios between technical replicates were averaged, while remaining separated for biological replicates. (iv) Only monophosphorylated peptides were considered for all subsequent analysis requiring quantifications at single positions. (v) To prevent the accumulation of false-positive sites, modified positions identified only in one single study were excluded. (vi) We quantile-normalized the quantifications across conditions and excluded all conditions with < 1,000 peptide quantifications. After applying all these quality control criteria, the size of the final set of conditions was reduced from 435 to 399 perturbations (Appendix Fig S1). On this reduced set of conditions, we considered for further analysis a total of 1,238,987 quantifications for 43,028 more reliable monophosphorylated peptides.

### Kinase set enrichment analysis

To estimate the kinase regulation from the differential regulation of their known substrates, we modified the original kinase set enrichment analysis to incorporate the principles of the weighted Gene Set Enrichment Analysis (Subramanian *et al*, 2005; Casado *et al*, 2013). This modified KSEA uses the Kolmogorov–Smirnov statistical test to assess whether a predefined set of kinase substrates is statistically enriched in phosphosites that are at the two of extremes of a ranked list defined by their differential regulation. This algorithm is particularly helpful to detect changes on phosphosite regulation in the

context of all site quantifications, even though the changes in site quantifications could be small. Moreover, the algorithm do not require any arbitrary threshold to define which phosphosites are significantly regulated or not. As in the GSEA algorithm, all site quantifications are considered in order to search for enrichments within the extreme fold changes. The KSEA algorithm proceeds as follows: (i) The Enrichment Score (ES) is calculated by walking down the ranked site quantifications and rescoring a running-sum statistic. The statistic is increased by the site quantification when it encounters a substrate of the kinase and decreases when the site is not a substrate. Both increases and decreases are normalized by the total number of known and not known substrates and proportional to the observed fold change. Finally, the ES corresponds to the maximum deviation from zero—either positive or negative—encountered in the walking down. The metric is equivalent to a weighted Kolmogorov–Smirnov-like statistic. (ii) The null distribution of ES is calculated by randomizing the sites but preserving the same distribution of site quantifications. (iii) The statistical significance of the observed ES is calculated using the empirical *P*-values calculated from the ES null distribution based on the same number of known substrates and the same distribution of quantifications. An R package implementing the novel KSEA described above is provided at https://github.com/evocellnet/ksea.

### Predicting kinase activities using KSEA

In order to collect a comprehensive list of regulatory relationships, we merged all the interactions reported in PhosphoSitePlus (Hornbeck *et al*, 2015), HPRD (Keshava *et al*, 2009) and Phospho.ELM (Dinkel *et al*, 2011) in June 2014. After excluding kinase auto-phosphorylations, we compiled a total of 7,815 interactions between 306 kinases and 5,617 individual sites. Using the above described regulatory information and the collected quantitative phosphoproteomic data, we applied the KSEA algorithm with a null distribution of 1,000 permutations per kinase and condition. In order to use the enrichment significance as a proxy of kinase activity, the resulting *P*-values were $\log_{10}$-transformed and signed based on the average sign of all substrates. If the predominant change of all substrates is an increase in phosphorylation, the kinase is predicted as activated. If the majority of the substrates present reduced phosphorylation, the kinase is predicted as inactivated.

### Kinase activity validation using kinase regulatory sites

To corroborate the kinase activity inference, KSEA activities were compared with the phosphorylation changes observed in kinase regulatory sites across conditions. From the total list of 941 human regulatory sites reported in PhosphoSitePlus, 150 were quantified in at least 10 perturbations. To evaluate the concordance between the KSEA activities and the quantitative changes in the regulatory phosphorylations, we performed a linear regression analysis across all available perturbations. Additionally, the results with those derived from the Spearman correlations between the kinase activities and the regulatory sites susceptible of auto-phosphorylation and other sites not reported to regulate the enzymatic activity. We discarded all kinase activity profiles under no regulation (absolute log *P*-value > 1 in at least 1 condition).

## Kinase activity validation using RPPA data

We compared the predictions based on the collected quantitative phosphoproteomic data with the antibody-based kinase activities from a previous study (Hill *et al*, 2016). We used the BT20 cell line as reference, showing the most responsive quantitative profiles after EGF receptor stimulation. We scaled the antibody-based measurements to make them comparable across conditions and antibodies. We quantile-normalized per antibody to assure equal final distributions. Next, we standardized each individual combination of cell line, inhibitor, stimulus, and time point by calculating the *z*-score of each of the measurements based on the mean and standard deviation of the unstimulated conditions. Replicates were averaged. The final dataset contains 26 quantifications reporting changes in regulatory phosphosites in kinases. In order to use only the most reliable activity predictions, kinases with a number of known substrates smaller than 5 were excluded. To circumvent the effect of protein abundances, we restricted the analysis to the first hour after EGF stimulation. The normalized quantifications clustered together based on the sample similarities, with no apparent batch effects (Appendix Fig S1). The DREAM conditions were classified depending if they activate EGFR—EGF and NRG1—other growth factors that eventually could have a similar downstream effects—HGF, IGF1, FGF1, and insulin—or non-stimulating conditions—serum and PBS.

## Maintenance and treatment of human embryonic stem cells (hESCs)

Human embryonic cells, H1 and H9 (WA01, WA09 from WiCell), were maintained on Matrigel (BD Biosciences)-coated dishes in mTeSR™1 medium (StemCell Technologies). Differentiation of hESCs was induced by supplementing mTeSR™1 with PMA 50 nM. Differentiation time course experiments were typically 0, 30 min, 1, 6 and 24 h. Kinase inhibition experiments were performed by supplementing mTeSR1 medium with pharmacological inhibitors 1 h before PMA treatments (Table EV8). In order to avoid inhibitor's degradation during 24-h experiment, fresh mTeSR1 medium with a 50 nM of PMA was changed after 10 h.

## hESC immunofluorescence and image analysis

For each time course experiment, hESCs were fixed for 10 min with a 4% paraformaldehyde, permeabilized for 5 min with 0.3% Triton X-100, and incubated with a blocking solution (10% fetal bovine serum (FBS) and 3% bovine serum albumin (BSA) in PBS) for 1 h. Primary antibodies (Table EV7) were incubated overnight at 4°C in antibody dilution buffer (1% bovine serum albumin (BSA), 0.1% Triton X-100 in PBS) at the indicated concentrations. Primary antibodies were visualized by using a secondary antibody conjugated to Alexa 488. Samples were counterstained with DAPI to facilitate analysis. Images were acquired using a high-content, widefield inverted microscope, Olympus ScanR System equipped with a sCMOS Flash 4.0 camera (Hamamatsu), universal plan semi-apochromat 20× objective (NA 0.7), and a SpectraX LED light source. Image analysis was performed using MATLAB or CellProfiler (Carpenter *et al*, 2006). Briefly, a low-pass Gaussian filter was first applied to each image. The local background value of each pixel was then determined by searching for a surrounding ring area,

with the outer and inner radii of the ring being 10 and 5 times the approximate nuclear radius, respectively. The lowest $5^{th}$ percentile value of the ring area was used as the background intensity of the center pixel. Cell nuclei were identified using fluorescent DAPI images as masks. When needed, cytoplasmic mask consisted of a ring around the nucleus. The MATLAB function *regionprops* was then used to label each nucleus and to retrieve the xy coordinates of all pixels in specific nuclei. The level of immunofluorescence staining in each cell was calculated as the average value of the intensities from each pixel of the specific nucleus. At least 2,000 cells were used for analysis per each indicated condition.

## PCA based on kinase activity profiles

To restrict the analysis to consistently estimated kinases, only those inferred in at least 75% of the perturbations were considered. Conditions displaying extreme redundancies were also excluded, reducing the matrix of kinase activities to 58 kinases and 387 conditions. For the 7.43% of the matrix containing missing values, the data were imputed using the regularized iterative PCA algorithm implemented in the *imputePCA* function contained in the R package *missMDA*. Using the resulting complete matrix, principal components analysis (PCA) was performed using the *rda* function in the R package *vegan* without any additional scaling. The expected (baseline) percent variance in each PC stemming from noise in data was estimated using the stringent "broken stick" method and the relaxed average eigenvalue (Kaiser–Guttman criterion) (Jackson, 1993).

## *Salmonella* strains used for infection

*Salmonella enterica* serovar Typhimurium 14028s (STm) transformed with the constitutive GPF expressing plasmid pDiGc (Helaine *et al*, 2010) were cultivated in LB broth (Miller) containing 100 μg/ml ampicillin by incubating on a rotating wheel at 37°C. HeLa cells (ATCC) were cultivated in DMEM 4.5 g/l glucose (Gibco cat. 41965-039), pyruvate (100 mM, Gibco), 10% FBS at 5% $CO_2$ in a 37°C incubator. Stock drug solutions were dissolved in DMSO: trichostatin A (Sigma cat. T8552) and SB202190 (Sigma cat. S7067), or methanol: (±)-verapamil hydrochloride (Sigma cat. V4629). Final drug concentrations used trichostatin A: 1.5, 1.0, and 0.5 μM; SB202190: 15, 10, and 5 μM; (±)-verapamil hydrochloride: 15, 10, and 5 μM. 100 mg/ml stock solution of gentamicin was dissolved in water (Sigma cat. G1914). Bacteria were prepared for HeLa cell invasion as previously described (Helaine *et al*, 2010) with the following modifications: Overnight cultures of GFP expressing STm were diluted 1:33 into fresh LB broth and cultured for 3.5 h at 37°C prior to infection.

## HeLa cell preparation and infection

At 80% confluency, 3,000 HeLa cells per well were seeded into a 384-well clear-bottom plate (Greiner cat. 781090) using a cell seeder (Thermo, Multidrop Combi) followed by an 18-h incubation overnight to allow cell attachment. Cells were then exposed to indicated drug concentrations in the presence of DMEM 1 g/l glucose + 10% FBS for 6 h. Prior to infection, cells were then washed two times with DMEM or PBS, followed by media

replacement with fresh DMEM 1 g/l glucose + 10% FBS. Infection was carried out as previously described [16] using a liquid handler (Biomek FXP). STm was added directly to HeLa cells at an MOI of 100 in PBS. STm was then allowed to invade HeLa cells by incubating for 30 min at 5% $CO_2$ in a 37°C incubator. Extracellular STm were then removed by washing three times with warm PBS, followed by treatment with 100 μg/ml gentamicin in DMEM 1 g/l glucose + 10% FBS for 1 h. Media were then replaced with 10 μg/ml gentamicin in DMEM 1 g/l glucose + 10% FBS for the remainder of the experiment n.b. This step was considered $t = 0$. At the indicated time points, cells were then washed with prewarmed PBS prior to fixation and permeabilization in 5% formaldehyde/0.2% Triton X-100 in PBS for 45 min. Fixing solution was then removed by washing with PBS and cells were stained using 2.5 μg/ml Hoechst33342 (Molecular Probes cat. H3570) and 80 ng/ml Phalloidin-Atto700 (Sigma cat. 79286) overnight at 4°C. Prior to imaging, cells were washed three times in PBS.

### *Salmonella* microscopy and image analysis

384-well plates were imaged using a Molecular Devices, IXM XL microscope where six sites per well were imaged at 20× magnification. CellProfiler was used to analyze the images. Nuclear regions were determined by setting a manual intensity threshold for the DAPI channel. Nuclei were expanded using the actin staining to determine the cellular regions. *Salmonella* colonies were determined by manual thresholding. Segmented cells were classified as infected or non-infected depending on the presence or the absence of a *Salmonella* colony in a cell region. For every site imaged, the number of infected and non-infected cells was determined, along with the integrated *Salmonella* fluorescence intensity inside infected cells. To determine the percentage of infected cells per well, the number of infected and non-infected cells from the six sites of every well was summed up and the ratio of infection was calculated. In addition, the mean integrated intensity of *Salmonella* in infected cells was determined for every site, and the average value for the six sites in a well was calculated to obtain the mean integrated intensity of *Salmonella* in infected cells per well as a measure of *Salmonella* intracellular proliferation.

### Co-regulation between driver kinases and effector complexes

We first quantified the phospho-regulation of stable human complexes [from the CORUM database (Ruepp *et al*, 2010)] in each condition. We limited the complex redundancy by subsetting the interactions in which only one copy of the homologous protein complexes is included. For each of the 1,331 complexes and for each condition, we compared the distribution of absolute changes in phosphosite abundance in the complex against all phosphosites using the Kolmogorov–Smirnov (KS) test. The resulting *P*-values were log-transformed and signed based on the average fold change of all sites in the complex. We then fitted a linear regression to estimate those responses in protein complex phosphorylation that correlate with changes in kinase activity across conditions. For validation purposes and in order to avoid potential biases, the kinase substrates used to predict the kinase activities were excluded from the complex regulation estimates. The Pearson correlation *P*-values were corrected for multiple testing.

### SILAC labeling, protein extraction, and digestion

HeLa cells were passaged in DMEM (-Arg, -Lys) with penicillin–streptomycin, 10% dialyzed FBS at 37°C, 5% $CO_2$, supplemented with either normal L-lysine and L-arginine (light K0, R0) or $^{13}C_6$-$^{15}N_2$ lysine and $^{13}C_6$-$^{15}N_4$ arginine (heavy K8, R10). Both populations of cells were deprived of serum overnight. "Light" labeled cells were treated with 1 μM Akt inhibitor VIII for 30 min prior to stimulation with 100 nM insulin for an additional 30 min. "Heavy" labeled cells were stimulated with 100 nM insulin for 30 min. At the time of harvest, cells were washed three times with ice-cold PBS and flash frozen over liquid nitrogen. Cells were scraped into ice-cold urea buffer (8 M urea, 75 mM NaCl, 50 mM Tris–HCl pH 8.2, complete protease inhibitor cocktail (Roche), 50 mM sodium fluoride, 50 mM beta-glycerophosphate, 1 mM sodium orthovanadate, 10 mM sodium pyrophosphate). Protein concentration was assayed using the BCA method and lysates from "light" and "heavy" cultures were mixed in a 1:1 ratio. Protein lysates were reduced with 5 mM DTT for 30 min at 55°C, alkylated with 10 mM iodoacetamide for 15 min at room temperature, and quenched with 10 mM DTT. Proteins were diluted twofold with 50 mM Tris pH 8.8 and digested with Lys-C (Wako) overnight at room temperature. The resulting peptides were desalted over a tC18 Sep-Pak cartridge (Waters) and dried by lyophilization.

### Strong cation exchange (SCX)/Immobilized metal affinity chromatography (IMAC)

Approximately 3 mg of peptides was resuspended in 50 mM Tris pH 8.2 and further digested with trypsin (Promega) overnight at 37°C. The resulting tryptic peptides were desalted over a C18 Sep-Pak cartridge (Waters) and dried by vacuum centrifugation. They were separated by strong cation exchange into 12 fractions using a volatile binary solvent system (A: 10 mM $NH_4HCO_2$ + 25% MeCN + 0.05% FA, B: 500 mM $NH_4HCO_2$ + 25% MeCN + 0.05% FA). Fractions were dried and desalted by vacuum centrifugation. Fractions were resuspended in 100 μl IMAC loading solution (80% MeCN + 0.1% TFA). To prepare IMAC slurry, Ni-NTA magnetic agarose (Qiagen) was stripped with 40 mM EDTA for 30 min, reloaded with 10 mM $FeCl_3$ for 30 min, washed three times, and resuspended in IMAC loading solution. To enrich phosphopeptides, 50 μl of 5% bead slurry was added to each fraction and incubated with rotation for 30 min at room temperature, washed three times with 150 μl 80% MeCN, 0.1% TFA, and eluted with 60 μl 1:1 MeCN:1% $NH_4OH$. The eluates were acidified with 10% FA and dried by vacuum centrifugation for LC-MS/MS.

### LC-MS/MS

Phosphopeptide-enriched samples were resuspended in 4% formic acid and 3% MeCN and subjected to liquid chromatography on an EASY-nLC II system equipped with a 100-μm inner diameter × 40 cm column packed in-house with Reprosil C18 1.9 μm particles (Dr. Maisch GmbH) and column oven set to 50°C. Separations were performed using gradients of 9–32% MeCN in 0.125% formic acid ranging in length from 55 to 105 min and were coupled directly with a LTQ-Orbitrap Velos mass spectrometer (Thermo Fisher) configured to conduct a full MS scan (60k resolution, 3e6 AGC target,

500 ms maximum injection time, 300–1,500 $m/z$) followed by up to 20 data-dependent MS/MS acquisitions on the top 20 most intense precursor ions (3e3 AGC target, 100 ms maximum injection time, 35% normalized collision energy, 40-s dynamic exclusion).

### LC-MS/MS data processing

Raw data files were converted to mzXML and searched using Comet version 2015.01 against the human SwissProt database including reviewed isoforms (April 2015; 42,121 entries) allowing for binary (all or none) labeling of lysine (+8.0142) and arginine (+10.0083), and variable oxidation of methionine, protein N-terminal acetylation, and phosphorylation of serine, threonine, and tyrosine residues. Carbamidomethylation of cysteines was set as a fixed modification. Trypsin (KR|P) fully digested was selected allowing for up to two missed cleavages. Precursor mass tolerance was set to 50 ppm and fragment ion tolerance to 1.0005 Daltons. Search results were filtered using Percolator to reach a 1% false discovery rate at the PSM level. Peak area heavy/light ratios were calculated using an in-house quantification algorithm. Phosphosite assignment was performed using an in-house implementation of Ascore, and sites with Ascore ≥ 13 were considered localized ($P = 0.05$). Phosphopeptides in the database with multiple non-localized instances spanning the same sequence were only considered to correspond to the minimum number of phosphosites that explain the data. Finally, the dataset was additionally filtered to reach a site-adjusted false discovery rate of 1%.

### Co-regulation between AKT and potential new substrates

For all human phosphorylated residues compiled in the atlas, a series of evidences were generated in order to weight their potential role as substrates of AKT. Firstly, all sites in the atlas were matched against the position weight matrixes (PWM) generated from all the known substrates of AKT. To weight the motif—PWM similarity, two algorithms were compared with similar results: the sum of the log weights of the matched residues and the MSS score provided by MATCH, which incorporates the information content of the PWM (Kel *et al*, 2003; Wasserman & Sandelin, 2004). Secondly, the aforementioned inference of kinase activities based on the known kinase substrates was used to find sites co-regulated with the known substrates of AKT. The response of all sites in the atlas across the panel of conditions was correlated with the estimated activity of AKT in the same conditions. We also included as an additional validation the set of 1,778 AKT *in vitro* substrates described by Imamura *et al* (2014). Finally, as a fourth independent evidence, we included the differential regulation of each of the candidate AKT sites under inhibition by AKT inhibitor VIII in insulin-stimulated HeLa cells described above.

**Expanded View** for this article is available online.

### Acknowledgements

We thank all the groups involved in generating the phosphoproteomics data for their invaluable public resource. We thank Jeff Johnson, Brandon Invergo, and Inigo Martincorena for critical reading of the manuscript, Omar Wagih and Francesco Ioirio for their help on implementing some of the methods, and Julio Saez-Rodríguez, John Marioni, and the rest of the group at the EMBL-EBI

for their insightful comments. PB acknowledges support from an HFSP CDA award (CDA00069/2013) and ERC Starting Grant (ERC-2014-STG 638884—PhosFunc). JV acknowledges support from a Howard Temin Pathway to Independence Award in Cancer Research (NIH K99/R00 CA140789) and an Ellison Medical Foundation New Scholar Award (AG-NS-0953-12).MJ and SDMS are supported by the Medical Research Council (MRC) (MCA652-5PZ600).

### Author contributions

DO, JV, SDMS, AT, and PB designed experiments. RTL, MJ, SDMS, JS, and BED performed the experiments. RL processed data the MS samples. DO performed the data analysis. DO and PB wrote the manuscript. PB oversaw the work. All authors read and approved the final manuscript.

### Conflict of interest

The authors declare that they have no conflict of interest.

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
