## [Review Process File · Molecular Systems Biology]

An Atlas of Human Kinase Regulation

David Ochoa, Mindaugas Jonikas, Robert Lawrence, Bachir El Debs, Joel Selkrig, Athanasios Typas, Judit Villen, Silvia Santos and Pedro Beltrao

Corresponding author: Pedro Beltrao, European Molecular Biology Laboratory - European Bioinformatics Institute

Review timeline:

Submission date:	30 August 2016
Editorial Decision:	27 September 2016
Revision received:	14 October 2016
Accepted:	20 October 2016

Editor: Thomas Lemberger

Transaction Report:

1st Editorial Decision

27 September 2016

Thank you again for submitting your work to Molecular Systems Biology. We have now heard back from the two referees who accepted to evaluate the study. As you will see, the referees find the topic of your study of potential interest and are supportive. They raise however a series of concerns and make suggestions for modifications, which we would ask you to carefully address in a revision of the present work.

- Reviewer #2 makes a series of suggestions to clarify some of the numbers presented and
- The presentation of the data and method of the results shown in Figure 2 should be improved; we attach an annotated PDF file to make some suggestions and comments.
- Please confirm that the download and mysql dump sections of the phosfate.com will be activated upon acceptance.
- We would be grateful if you could complete Table S1 with the accession numbers and resource of the dataset, at least for those 16 studies where the data have been deposited. If supplementary files from the respective papers have been used, it might be convenient to provide the file/table number or name as well as the URL.

REFeree REPORTS

Reviewer #1:

Ochoa, et al.

I reviewed this manuscript previously and had strongly recommended it for publication. I also see that they had integrated corrections in response to mine and a number of other issues that the other reviewer had raised. My opinion has not changed, but nonetheless, here are my original comments.

The title of this work is apt, for it seems the focus of the authors efforts is to order the regulation of protein kinases according to cellular functions and to each other in the context of different cell functions. What Ochoa, et al. describe seems to be straightforward. They have compiled just about all of the human condition-specific differential phosphoproteomic data that they could muster, matched the phosphosites to those of known substrates of kinases and were able to come up with a compendium of condition specific activation states for 209 human protein kinases, about half of the human kinome. This is very good indeed. What follows then is a series of analyses in which the authors explore what this large scale knowledge of what states kinases are in under such a large number of conditions might tell us about the nature of individual kinases, their relationships to other kinases and to cellular functions. Ultimately what we're provided with here is a collection of virtual reporters of signaling network regulation. To demonstrate the utility of these virtual reporters, the authors analyzed responses of different kinases during PMA-driven differential of hESC cells using phospho-substrate-specific antibodies as reporters to test the predictive power of their method to associate the regulation of different kinases to distinct conditions.

The analyses described in this manuscript are quite elegant, straight-forward and answer some of the more basic questions one should ask given our present understanding that kinases form regulatory signaling networks that are intrinsically complex, limiting ourselves at the moment to describing kinase activity perturbations under different conditions as signatures of those conditions that may have some considerable predictive utility. This is a major contribution that the authors have done such a large-scale analysis and should be of general interest and utility to those interested in signaling networks.

Reviewer #2:

The authors have compiled a large data collection of almost 3 million human phosphopeptide quantifications from 435 perturbations. They collected these based on existing data, renormalized and filtered them. Kinase activities were inferred from the quantifications and show good correlation with autophosphorylation signals. Generalist kinases are identified that are central in the signaling network and pass through signals to other kinases. The study is unique and of excellent quality, the results support the conclusions. However I find the presentation of numbers included in the study confusing and in places misleading. I understand that the authors try to represent the large amount of analytical work that has gone into this study, but as it is, it is difficult to read and understand the results. Although breakthrough results are missing, the Atlas could represent a very valuable resource for the PTM community.

Presentation of numbers

The abstract does not mention how many phosphopeptides actually change and are thus part of a profile. My first impression was that the 3 million represented phosphopeptide changes (instead of quantifications). The way it is presented now is confusing. Further, the authors report that in fact only 48% of these 3 million quantifications are retained as the rest concerns phosphopeptides only seen in one study and are likely false positives.

Similarly, on page 4 is written "In this study, we have compiled condition-dependent changes in

human protein phosphorylation including 2,940,379 phosphopeptide quantifications." The word "including" seems to suggest all those quantified peptides are changed. It should rather be worded as "based on" or "derived from".

Please include the number of changes in phosphopeptides in this sentence as well as the number of unique phosphopeptides that have been quantified and display significant changes. Only in the supplement it is clear that only about 100,000 unique phosphopeptides have been quantified. How many of those end up in the final profiles is unclear.

Also the abstract mentions 399 conditions whereas on page 4, 435 perturbations are mentioned. Where does this difference come from? Do the other 36 conditions only contain phosphopeptides that are unique to one condition? Or do they not belong to a kinase?

Page 12:

The authors present correlation as R in the text and the figure. It should rather be presented as R² which is the conventional way. This is again confusing and could be interpreted as misleading (a reader expects to see the R-squared here). Similarly on page 13 (R=0.27).

It is unclear how the correlation in figure 4C is calculated. Is this Pearson correlation? Are the underlying assumptions for Pearson correlation fulfilled? That is, are the two variables normally distributed? Would a rank correlation not be more appropriate?

Page 13:

"Protein complexes are common signaling effectors that often display coordinated phospho-regulation with regulatory kinases" Is this an assumption? Is it based on previously published results? Is this a result of the study? Please clarify.

Fig S3

For correlation between kinase activity profiles, Pearson correlation is taken. Are the profiles normally distributed? Is a Pearson correlation justified here? Should it not rather be a rank correlation?

Minor

Page 3: "improvements ... has fostered". ⇒ "have fostered" (plural)

Dear Thomas,

Please find below our point by point response to your comments and the comments of the 2 reviewers. Our response is in black. We have made all minor changes requested as well as the changes to the figure numbering, supplementary materials

Dear Pedro,

Thank you again for submitting your work to Molecular Systems Biology. We have now heard back from the two referees who accepted to evaluate the study. As you will see, the referees find the topic of your study of potential interest and are supportive. They raise however a series of concerns and make suggestions for modifications, which we would ask you to carefully address in a revision of the present work.

- Reviewer #2 makes a series of suggestions to clarify some of the numbers presented and

- The presentation of the data and method of the results shown in Figure 2 should be improved; we attach an annotated PDF file to make some suggestions and comments.

After critical reading of the section, we understand the concerns and we made a number of changes to clarify the message. The results as well as the scientific interpretation remain the same, but we've tried to clarify the way we presented the results.

We rewrote a few sentences in the section related to Figure 2 and clarified some others. In particular, we tried to make more transparent the logic of why the PMA experiment was made and the usefulness of the substrate-based kinase activity prediction to answer specific biological questions. We also tried to explain better how the KSEA results help to identify functionally relevant kinases and why this suggests that the atlas is a useful cell signaling resource.

We updated Figure 2C as the length and color of the bars contained redundant information. By removing the colors the interpretation of the bars in the context of the legend is more intuitive.

To clarify the meaning of the horizontal bars in Figure EV2 (formerly Figure S4), we added the next sentence to the legend:

*"...The KSEA activities are inferred from the MS-quantified substrates in the time intervals 0-30 and 0-60 minutes after PMA stimulation and represented as a fraction of the theoretical limit based on the number of KSEA permutations. **Note that the MS-based horizontal bars represent a time interval (i.e. 0-30min) as compared to the IHC data measured in each individual timepoint (i.e. 30 min)...**"*

- Please confirm that the download and mysql dump sections of the phosphate.com will be activated upon acceptance.

We already activated the "Download" section in <http://phosphate.com>. This page contains the mysql dump with all standardized human phosphopeptide quantifications obtained from 41 different publications, as well as metadata about the experimental conditions and biological samples. We also include some instructions on how to restore the data to have full access to the database. On the same web page, the

community can access some of the inferences described in the manuscript, such as the kinase activities or the kinase-complex associations.

- We would be grateful if you could complete Table S1 with the accession numbers and resource of the dataset, at least for those 16 studies where the data have been deposited. If supplementary files from the respective papers have been used, it might be convenient to provide the file/table number or name as well as the URL.

We updated Table S1 to contain the raw spectra database and id for the the 16 publicly available datasets. For the particular case of the datasets deposited in Tranche, we clarified the database is “discontinued” as the data is no longer available. To complement the pubmed ids, we included the url of the article based on the more stable doi identifiers (<https://www.doi.org>) rather than in the journal websites.

- As you may have noticed, we recently replaced Supplementary Information by Expanded View (EV, see examples in <http://msb.embopress.org/content/11/6/812>). In this format, a limited number of Supplementary Figures (max 5) can be integrated in the article as EV figures that are interactively collapsible/expandable and will be typeset by the publisher. In this case, the figures should be cited as 'Figure EV1, Figure EV2" etc... in the text and their respective legends should be added to the main text after the legends of regular figures. The illustrations should be provided as separate files.

To comply with the new MSB author guidelines, we included 5 figures previously labelled as “Supplementary Figures” as “Expanded View (EV) Figures”. Their respective legends remain unchanged and can be found in the main manuscript after the legends of the main figures. The references to the figures in the text were modified accordingly.

- For the figures that you do NOT wish to display as Expanded View figures items, they should be bundled together with their legends in a 'traditional' supplementary PDF, now called the *Appendix*. Appendix should start with a short Table of Content and the figures should be named and referred to in the main text as: "Appendix Figure S1, Appendix Figure S2" etc. See detailed instructions regarding expanded view here: <http://msb.embopress.org/authorguide#expandedview>.

The remaining supplementary figures not included in the “Expanded View Figures” were included in the “Appendix” pdf and renamed to “Appendix Figure SX”. The “Appendix” pdf starts with a table of contents enumerating the figures contained in the file. All “Appendix Figures” were referenced in the main text.

- Additional Tables/Datasets should be labeled and referred to as Table (or Dataset) EV1 etc. Table/Dataset legends can be provided in a separate tab in case of .xls files. Alternatively, you can upload a .zip file containing the Table/Dataset file and a separate README .txt file with the legend/description.

The “Supplementary Tables” were renamed to “Expanded View Tables” and referenced using “Table EVX”. Each table was wrapped in a zip file containing the Microsoft Excel file and a README.txt with the legend/description.

Please resubmit your revised manuscript online, with a covering letter listing amendments and responses to each point raised by the referees. Please resubmit the paper *within one month*******

and ideally as soon as possible. If we do not receive the revised manuscript within this time period, the file might be closed and any subsequent resubmission would be treated as a new manuscript. Please use the Manuscript Number (above) in all correspondence.

Reviewer #1:

Ochoa, et al.

I reviewed this manuscript previously and had strongly recommended it for publication. I also see that they had integrated corrections in response to mine and a number of other issues that the other reviewer had raised. My opinion has not changed, but nonetheless, here are my original comments.

The title of this work is apt, for it seems the focus of the authors efforts is to order the regulation of protein kinases according to cellular functions and to each other in the context of different cell functions. What Ochoa, et al. describe seems to be straightforward. They have compiled just about all of the human condition-specific differential phosphoproteomic data that they could muster, matched the phosphosites to those of known substrates of kinases and were able to come up with a compendium of condition specific activation states for 209 human protein kinases, about half of the human kinome. This is very good indeed What follows then is a series of analyses in which the authors explore what this large scale knowledge of what states kinases are in under such a large number of conditions might tell us about the nature of individual kinases, their relationships to other kinases and to cellular functions. Ultimately what we're provided with here is a collection of virtual reporters of signaling network regulation. To demonstrate the utility of these virtual reporters, the authors analyzed responses of different kinases during PMA-driven differential of hESC cells using phospho-substrate-specific antibodies as reporters to test the predictive power of their method to associate the regulation of different kinases to distinct conditions.

The analyses described in this manuscript are quite elegant, straight-forward and answer some of the more basic questions one should ask given our present understanding that kinases form regulatory signaling networks that are intrinsically complex, limiting ourselves at the moment to describing kinase activity perturbations under different conditions as signatures of those conditions that may have some considerable predictive utility. This is a major contribution that the authors have done such a large-scale analysis and should be of general interest and utility to those interested in signaling networks.

We thank the reviewer for the very positive comments and interest in our work.

Reviewer #2:

The authors have compiled a large data collection of almost 3 million human phosphopeptide quantifications from 435 perturbations. They collected these based on existing data, renormalized and filtered them. Kinase activities were inferred from the quantifications and show good correlation with autophosphorylation signals. Generalist kinases are identified that are central in

the signaling network and pass through signals to other kinases. The study is unique and of excellent quality, the results support the conclusions. However I find the presentation of numbers included in the study confusing and in places misleading. I understand that the authors try to represent the large amount of analytical work that has gone into this study, but as it is, it is difficult to read and understand the results. Although breakthrough results are missing, the Atlas could represent a very valuable resource for the PTM community.

Presentation of numbers

The abstract does not mention how many phosphopeptides actually change and are thus part of a profile. My first impression was that the 3 million represented phosphopeptide changes (instead of quantifications). The way it is presented now is confusing. Further, the authors report that in fact only 48% of these 3 million quantifications are retained as the rest concerns phosphopeptides only seen in one study and are likely false positives.

Similarly, on page 4 is written "In this study, we have compiled condition-dependent changes in human protein phosphorylation including 2,940,379 phosphopeptide quantifications." The word "including" seems to suggest all those quantified peptides are changed. It should rather be worded as "based on" or "derived from".

Because of the reviewer's confusion with the numbers, we have made an effort to complete and clarify some of the raised points that might not be straight-forward to all readers. One of the problems seems to be derived with the ambiguity when referring to "phosphopeptide quantifications". Whereas we referred to the quantified fold changes between a given perturbation and the control, the reviewer understood that this could refer only to phosphosites that change significantly. We included a few modifications in the text, so this concept can be more quickly assimilated by the reader.

In the abstract, we mentioned:

"Here, we have estimated changes in activity in 215 human kinases in 399 conditions from a compilation of nearly 3 million phosphopeptide quantifications."

As the reviewer found the 3 million value in the abstract potentially misleading we have removed it:

*"Here, we have estimated changes in activity in 215 human kinases in 399 conditions **derived from a large compilation of phosphopeptide quantifications.**"*

As suggested by the reviewer, we've also changed the "Introduction" section to clarify the same point.

*"In this study, we have compiled condition-dependent changes in human protein phosphorylation **including derived from** 2,940,379 phosphopeptide quantifications in 435 heterogeneous perturbations."*

We don't think the concept requires more clarification in the results section where we refer precisely to "quantitative changes":

"For these sites, we normalized a total of 2,940,379 quantitative changes in phosphopeptide abundance in a panel of 435 biological conditions covering a broad spectrum of perturbations"

In the "Materials and Methods", where we spend more space explaining how the data was collected, we explain that this number represents the total amount of quantified sites across all experiments:

*“For these reasons we relied on the original MS identifications performed by the experimental groups. We collected all quantifications from the supplementary data available in each of the 41 studies, **including both significant and not significant changes.**”*

Another point raised by the reviewer is that we only retained the 48% of the ~3M sites that show up in more than 1 experiment. We want to clarify that even if we use this threshold to prevent the aggregation of false positives, the chances that many relevant phosphorylation sites can be found within the remaining 52% are high. For this reason, we consider that the full atlas containing the 3M quantification might be of interest for the community.

Please include the number of changes in phosphopeptides in this sentence as well as the number of unique phosphopeptides that have been quantified and display significant changes. Only in the supplement it is clear that only about 100,000 unique phosphopeptides have been quantified. How many of those end up in the final profiles is unclear.

A total of 43.028 monophosphorylated phosphopeptides were quantified in at least 2 different publications. For these phosphosites we compiled a total of 1.238.987 measurements in the 399 perturbations that passed the quality control criteria. These numbers were clarified in the “Materials & Methods” section:

“On this reduced set of conditions, we considered for further analysis a total of 1.238.987 quantifications for 43.028 more reliable monophosphorylated peptides.”

It is important to note that we do not use a statistical threshold to decide individual sites that are changing significantly. Due to the different nature of the compiled experiments, setting up a common significance threshold would assume that the fold-change differences across conditions would have the same statistical meaning. Additionally, the correction for multiple testing would require an additional layer of complexity. Instead, we evaluated sets of phosphopeptide quantifications relative to the changes of the whole profile of normalized quantifications within the same experiment. We take advantage of statistical methods such as the GSEA or the KS tests that do not require the categorization of each individual site as significantly regulated or not. As a consequence of using this methods, the whole profile of quantified sites in any given perturbation is used to evaluate the activity of each kinase or regulation of each complex. Therefore, all 1.238.987 measurements are actively used for the inferences.

To clarify some of these points, the next sentences were added to the “Kinase Set Enrichment Analysis (KSEA)” in the “Materials and Methods” section:

*“This algorithm is particularly helpful to detect changes on phosphosite regulation in the context of all site quantifications, even though the changes in site quantifications could be small. **Moreover, the algorithm do not require any arbitrary threshold to define which phosphosites are significantly regulated or not. As in the GSEA algorithm, all site quantifications are considered in order to search for enrichments within the extreme fold changes.** The KSEA algorithm proceeds as follows:”*

Also the abstract mentions 399 conditions whereas on page 4, 435 perturbations are mentioned. Where does this difference come from? Do the other 36 conditions only contain phosphopeptides that are unique to one condition? Or do they not belong to a kinase?

We agree with the reviewer that due to the different quality control steps some of the numbers might not have been clarified enough in the main text. The reduction from the 435 conditions collected and the 399 conditions finally used in the kinase activity predictions is due to the different filtering steps. As the

reviewer pointed out and we described in the “Materials and Methods” section, all phosphosites must be in more than 1 condition to be considered for further analysis. Other filter requires only monophosphorylated sites to be used in the kinase activity predictions. As consequence of each of these steps, the number of phosphopeptides might be reduced significantly in each of the conditions. To be sure enough phospho-regulatory data is contained in each condition, only those perturbations with more than 1,000 quantified monophosphorylated phosphopeptides were considered, reducing the set from 435 conditions to 399. To make this point more clear on the main text we added the next words in bold:

*“In this study, we have compiled condition-dependent changes in human protein phosphorylation including 2,940,379 phosphopeptide quantifications in 435 heterogeneous perturbations. After quality control and normalization, we infer and benchmark the changes in 215 kinase activities **in 399 conditions.**”*

In the “Materials and Methods” section, we’ve expanded the explanation:

(original)

“(f) we quantile normalized the quantifications across conditions and excluded all conditions with less than 1000 peptide quantifications to the final 399 perturbations (Fig. S3).”

(modified)

*“(f) we quantile normalized the quantifications across conditions and excluded all conditions with less than 1000 peptide quantifications. **After applying all these quality control criteria, the size of the final set of conditions was reduced from the 435 to 399 perturbations (Appendix Figure S1).**”*

We also considered to remove the 36 discarded conditions from the atlas. However, we thought a larger compendium might represent a more useful resource for the community.

Page 12:

The authors present correlation as R in the tekst and the figure. It should rather be presented as R^2 which is the conventional way. This is again confusing and could be interpreted as misleading (a reader expects to see the R -squared here). Similarly on page 13 ($R=0.27$).

It is unclear how the correlation in figure 4C is calculated. Is this pearson correlation? Are the underlying assumptions for pearson correlation fulfilled? That is, are the two variables normally distributed? Would a rank correlation not be more appropriate?

Considering we cannot validate the underlying data to be normally distributed, we opted to follow the reviewer recommendations and modify the metrics to the more conservative Spearman’s rank correlation. Therefore, Figure 1D-G, Figure 4C and Appendix Figure S2 were modified, as well as their figure legends, main text and methods. After changing the statistical method to Spearman none of the hypothesis tested was affected and all the conclusions remained the same. In order to reduce the ambiguity, we specified the method when referring to correlations. As proposed by the reviewer, we also replaced the Pearson correlation coefficient (r) with the Spearman Rank coefficient (ρ) in Figure 1D, Figure 1F and Figure 4C.

Page 13:

“Protein complexes are common signaling effectors that often display coordinated

phospho-regulation with regulatory kinases" Is this an assumption? Is it based on previously published results? Is this a result of the study? Please clarify.

We agree with the reviewer that this sentence was not sufficiently clear. We made the assumption that phosphosites within proteins of a complex could be coordinately regulated and that this could serve as a signal for the regulation of the complex itself. Under this assumption we used a statistical test to identify, in each condition, which complexes show a significant change in their phosphorylation state. We have updated this sentence in the manuscript to include this assumption:

"We hypothesize that effector protein complexes would display coordinated phospho-regulation with regulatory kinases."

Fig S3

For correlation between kinase activity profiles, pearson correlation is taken. Are the profiles normally distributed? Is a pearson correlation justified here? Should it not rather be a rank correlation?

This figure was updated with the Spearman correlation as discussed above. We appreciate the comment.

⇒

Minor

Page 3: "improvements ... has fostered". "have fostered" (plural)

Typo fixed.

Thank you again for sending us your revised manuscript. We are now satisfied with the modifications made and I am pleased to inform you that your paper has been accepted for publication.

Corresponding Author Name: Pedro Beltrao

Manuscript Number: MSB-16-7295